# Concentrations of N^6^-Carboxymethyllysine (CML), N^6^-Carboxyethyllysine (CEL), and Soluble Receptor for Advanced Glycation End-Products (sRAGE) Are Increased in Psoriatic Patients

**DOI:** 10.3390/biom12121870

**Published:** 2022-12-13

**Authors:** Aleksandra Damasiewicz-Bodzek, Agnieszka Nowak

**Affiliations:** Department of Chemistry, Faculty of Medical Sciences in Zabrze, Medical University of Silesia, 40-055 Katowice, Poland

**Keywords:** psoriasis, CML, CEL, sRAGE, glycoxidation

## Abstract

Psoriasis is a chronic, recurrent, and often severe skin disease which is frequently associated with metabolic disorders and increased risk of cardiovascular complications. One of the postulated links is an intensified process of advanced protein glycation and/or glycoxidation. Therefore, the aim of the study was to assess concentrations of N^6^-carboxymethyllysine (CML), N^6^-carboxyethyllysine (CEL), and soluble form of receptor for advanced glycation end-products (sRAGE) in psoriasis patients at different phases of the disease activity, in comparison to healthy individuals. The study material consisted of sera from psoriasis patients in active phase, in the remission phase, and healthy controls. Concentrations of CML, CEL, and sRAGE were determined using ELISA technique. In the patients with psoriasis (in both phases of the disease), concentrations of CML, CEL and sRAGE were significantly higher than in healthy individuals but they did not correlate with psoriasis area severity index (PASI) values. The remission of the disease was followed by a significant decrease in CML, CEL, and sRAGE concentrations when compared to active patients; however, these concentrations were still significantly higher than in the controls. Our data suggest that psoriasis is accompanied by an intense glycoxidation process and that high sRAGE levels seem to reflect permanent RAGE overstimulation.

## 1. Introduction

Psoriasis is a chronic, recurrent, and, in many cases, severe skin disease that significantly reduces patients’ quality of life. It affects 2–3% of the western countries’ population [1]. Of the patients with psoriasis, 10–20% have involved body surface area greater than 10%, psoriasis of the scalp develops in 75–90% of cases, and nail psoriasis occurs in about 50% of patients. Additionally, 30% of patients suffer from psoriatic arthritis [2]. The essence of the disease is accelerated and abnormal keratinization of epidermal cells, accompanied by an inflammatory infiltration in the dermis. The complex etiopathogenesis of psoriasis causes serious therapeutic difficulties despite the existence of many methods of treatment. Genetic, vascular, inflammatory, infectious, immunological, autoimmunological, hormonal, psychosomatic, environmental, and other factors are important in the development of this dermatosis. An additional health problem is the fact that psoriasis is frequently associated with metabolic and lipid disorders, including obesity, type II diabetes, dyslipidemia, atherosclerosis, arterial hypertension, and increased risk of cardiovascular complications [3,4,5,6,7,8,9,10,11,12,13,14]. However, the precise mechanism connecting these diseases with psoriasis is still unknown. One of the postulated links is an intensified process of advanced protein glycation and/or glycoxidation.

Glycation is a complex network of interdependent reactions. It is based on a nucleophilic attack of an amino group (of a free or peptide-bound amino acid) on a carbonyl compound (for example monosaccharide, glyoxal, methylglyoxal) [15]. Products of the reaction may be divided into early glycation products (EGPs), intermediate glycation products (IGPs), and advanced glycation end-products (AGEs) [16]. Examples of AGEs include N^6^-carboxymethyllysine (CML), N^6^-carboxyethyllysine (CEL), N^7^-carboxymethylarginine (CMA), N^7^-carboxyethylarginine (CEA), pentosidine, glucosepane, and many more [17]. Some of the glycation reactions require oxidative conditions, and oxidation contributes to formation of some glycation substrates (e.g., glyoxal). Therefore, glycation is tightly connected to oxidative stress, and is sometimes termed glycoxidation [17,18]. AGEs are formed under physiological and pathological conditions. It is known that total amount of AGEs increases with age and during chronic diseases [18]. AGEs affect functions of cells and extracellular matrix. Compounds such as pentosidine and glucosepane cross-link proteins. Some AGEs (glyceraldehyde derivatives) are considered cytotoxic. AGEs are ligands of proinflammatory receptor for advanced glycation end-products (RAGE). There is a soluble form of this receptor (sRAGE) that acts anti-inflammatory, as it is a scavenger receptor for RAGE’s ligands [19,20,21].

Glycation/glycoxidation processes may constitute a potential bridge between psoriasis and increased incidence of diabetes, atherosclerosis, and cardiovascular diseases. Therefore, the aim of our study was to assess the concentration of CML, CEL, and sRAGE in sera of psoriatic patients at two different stages of the disease activity (active phase and remission phase), in comparison to samples obtained from healthy individuals.

## 2. Materials and Methods

### 2.1. Study Participants

Archival samples of serum stored in deep freezing were used in the study. The tested group included 40 patients with psoriasis (20 women and 20 men, mean age: 37.2 ± 11.5 years) admitted to a dermatological ward due to intense skin lesions. Mean disease duration time was 9.7 ± 7.0 years (range 1–276 months). The blood samples were drawn for the first time in the active phase (mean psoriasis area severity index PASI: 27 ± 14) before any anti-psoriasis treatment. Blood was drawn for the second time during the remission phase. The remission criteria were PASI value below 3 or the value reduced by 90% or more. Remission was obtained by different therapeutic methods: 11 patients were treated with topical corticosteroids and phototherapy (narrow band UVB; 311 nm), 14 patients were treated with acitretin (a second-generation retinoid), and the remaining 15 were treated with ciclosporin (an immunosuppressive drug). The mean time from the beginning of the treatment to the disease remission was 6.2 ± 5.5 weeks (range 2–24 weeks). Any concurrent diseases were considered exclusion criteria. To perform a detailed data analysis, patients in the active phase were divided according to PASI values and disease duration time. Patients in the remission phase were divided according to therapeutic method and time to remission (Table 1).

The control group included 35 healthy volunteers (17 women and 18 men), at comparable age (33.8 ± 9.2 years; *p* > 0.05), with no cases of psoriasis in their families. Local Bioethical Commission of Silesian Medical University in Katowice accepted the study protocol. All participants signed informed consent before participating in this study.

### 2.2. Samples and Assays

Blood samples were drawn on fasting and left to clot. Sera obtained by centrifugation was stored at −80 °C until analysis was performed. Enzyme-linked immunosorbent assay (ELISA) technique was used to evaluate concentrations of selected parameters of advanced protein glycation in tested serum samples. Following commercially available kits were used: OxiSelect N^ε^-(carboxymethyl)lysine (CML) Competitive ELISA Kit, OxiSelect N^ε^-(carboxyethyl)lysine (CEL) Competitive ELISA Kit (both from Cell Biolabs Inc., San Diego, CA, USA), and RayBio^®^ Human RAGE ELISA Kit (RayBiotech, Inc., Peachtree Corners, GA, USA). These kits were used to determine CML, CEL, and sRAGE, respectively.

Concentrations of CML and CEL were estimated by applying competitive ELISA technique, following similar protocols. First, 96-well plates were coated with CML and CEL conjugates, respectively. On the next day (after overnight incubation in 4 °C), CML/CEL standards and serum samples were pipetted into to the wells in duplicates. After 10-min incubation, anti-CML/anti-CEL monoclonal antibodies were added too. Then, plates were incubated for 1 h and washed 3 times. Next, a conjugate of secondary antibody and horseradish peroxidase was added. The plates were incubated for 1 h and washed 3 times again. Afterwards, substrate solution was added. The plates were incubated for 15 min, based on the color development. Stop solution was added to stop the reaction and absorbance measurement was performed immediately.

Concentrations of sRAGE were estimated by applying sandwich ELISA technique. 96-well plates were pre-coated with human RAGE antibodies. First standards and serum samples were added into the wells. The plates were incubated for 2.5 h and washed 4 times. Then, biotinylated antibody was added and the plates were incubated for 1 h and washed 4 times again. Afterwards, streptavidin solution was pipetted into the wells. The plates were incubated for 45 min and washed 4 times again. 3,3′,5,5′-tetramethylbenzidine was added and the plates were incubated for 30 min. Stop solution was added to stop the reaction and absorbance was measured immediately.

Detailed protocols are available at manufacturers websites. To perform washes, 50 TS microplate washer (BioTek Instruments, Inc., Winooski, VT, USA) was used. ELMI DTS-2 shaker-thermostat (ELMI Ltd., Riga, Latvia) was used to incubate and shake the plates. Absorbances were measured with Power Wave XS plate reader (BioTek, Winooski, VT, USA) at 450 nm (reference wavelength 630 nm), and results were calculated with KC Junior software (BioTek, Winooski, VT, USA). All samples were analyzed in one series. The intra-assay variation was below 8%, sensitivity was 2.25 ng/mL for CML, 0.1 μg/mL for CEL, and 3 pg/mL for sRAGE.

### 2.3. Statistics

Basic descriptive statistics parameters (mean/median and standard deviation) were used to present the results. Shapiro–Wilk’s test was applied to estimate if distribution of the data was normal. Non-parametric Kołmogorow–Smirnow and *U* Mann–Whitney tests were used to compare independent data between the groups of subjects with psoriasis (in both stages of the disease) and the control group. Wilcoxon’s pair test was used to compare dependent data between the active stage and the remission stage of the disease. Analysis of variance (ANOVA) and Kruskal–Wallis test were used to compare the results within the groups. Multivariate analysis of variance (MANOVA) was used to evaluate combined influence of independent variables on dependent variables. The *p* < 0.05 was considered as statistically significant. STATISTICA for Windows 13.3 software (TIBCO Software Inc., Palo Alto, CA, USA) was used to perform the aforementioned calculations.

## 3. Results

The results of measurements are presented in Table 2 and illustrated in Figure 1.

Analysis of the obtained data showed that in the patients with active psoriasis the mean concentrations of CML, CEL, and sRAGE were significantly higher than in the healthy controls (*p* < 0.000001; *p* < 0.0005; *p* < 0.000001, respectively). The remission of the disease was followed by a statistically significant decrease in CML, CEL, and sRAGE concentrations, when compared to active patients (*p* < 0.0005; *p* < 0.01; *p* < 0.005, respectively). However, the mean concentration of these parameters was still significantly higher than in the controls (*p* < 0.00005; *p* < 0.01; *p* < 0.00005, respectively).

The serum CML, CEL, and sRAGE levels did not correlate with age, both in the psoriasis group and in the healthy controls. The correlations of examined parameters are presented in Table 3 and the correlations between their values in active and remission phase are illustrated in Figure 2. None of the examined parameters correlated with PASI values, time of disease duration, or time from the beginning of treatment to the disease remission.

In the active phase of the disease mean concentrations of examined parameters (CML, CEL, sRAGE) did not differ significantly between the PASI value subgroups (*p* = 0.65; *p* = 0.11; *p* = 0.51, respectively) and between the disease duration time subgroups (*p* = 0.91; *p* = 0.55; *p* = 0.53, respectively).

In the remission phase, mean concentrations of CML and sRAGE did not differ significantly between the therapeutic method subgroups (*p* = 0.49; *p* = 0.18, respectively) and the time to remission subgroups (*p* = 0.58; *p* = 0.17, respectively). Interestingly, during remission mean concentration of CEL was significantly higher in the patients treated with acitretin than in the patients treated with topical corticosteroids and UVB phototherapy (523.23 ± 174.21 ng/mL vs. 368.49 ± 142.15 ng/mL; *p* < 0.05). No difference in mean concentration of CEL between the time to remission subgroups was observed (*p* = 0.86).

Multivariate analysis did not show a combined influence of PASI value and time of the disease on mean concentrations of CML (*p* = 0.77), CEL (*p* = 0.52), and sRAGE (*p* = 0.68) during the active phase. There was also no combined influence of therapeutical method and time to remission on mean concentrations of CML (*p* = 0.28) and sRAGE (*p* = 0.86) in the remission phase. However, there was a combined influence of therapeutical method and time to remission on CEL concentration in the remission phase (*p* < 0.05). Patients treated with acitretin for more than 6 weeks have significantly higher mean concentrations of CEL than patients in all other groups.

## 4. Discussion

AGEs are generated in vivo, under physiological and pathological conditions [22]. Among pathological conditions known to intensify the process of glycation there are oxidative stress, carbonyl stress, hyperglycemia, hypoxia, and chronic inflammation. Additionally, there are exogenous sources of AGEs, such as diet, UV irradiation, and tobacco smoke [23,24,25]. AGEs are heavily involved in pathophysiology and course of inflammatory and metabolic disorders by increasing cytokines and chemokines release, free radicals production, and metalloproteases activation [25]. AGEs bind to pro-inflammatory receptor RAGE, and thereby activate transcription factor NF-κB signaling pathway. NF-κB is able to interact with promoter of the RAGE gene itself and regulate its expression. RAGE expression is upregulated in areas rich in its ligands. Therefore, there is a positive feedback loop between the processes of glycation and inflammation [20,26].

Psoriasis is an immune-mediated, chronic inflammatory disease that increases risk of cardiovascular morbidity [27,28]. It may be hypothesized that accumulation of AGEs related to chronic psoriatic inflammatory state is a potential link between psoriasis, accelerated atherosclerosis, and its complications. It was proven that accumulation of AGEs in psoriatic skin (evaluated as skin autofluorescence) is significantly higher in patients with intensified inflammation (measured as higher levels of C-reactive protein) and prediabetes (HbA1c 5.7–6.4%) [29]. Additionally, skin autofluorescence correlates positively with carotid intima-media thickness (IMT) (atherosclerosis and cardiovascular events risk marker) [24].

CML and CEL are the most representative compounds used to evaluate the extent of glycation/glycoxidation and lipooxidation processes. CML and CEL are formed via nucleophilic addition of lysine to glyoxal and methylglyoxal, respectively [30].

The analysis of data obtained in the study proved that psoriasis is accompanied by an increased accumulation of CML and CEL in blood serum of the patients. These results agree with previously observed increased levels of methylglyoxal (CEL precursor) [31], anti-CML and anti-CEL antibodies [32] in psoriasis. Skin is a tissue that is very sensitive to changes induced by AGEs, because it contains long-lived proteins. AGEs accumulation in the skin enhanced production of free radicals what, in consequence, led to intensified peroxidation processes [33,34]. Dermal fibroblasts in psoriasis (with and without lesions) exhibit increased levels of carbonyl residues, as evidence of oxidative damage [35]. The oxidative damage and the increased level of methylglyoxal worsen the course of the disease, there is a significant correlation between serum levels of methylglyoxal and psoriasis severity [31]. It was proven that AGEs concentration in blood correlates strongly with their concentration in skin [36]. Therefore, it could be hypothesized that high concentrations of CML and CEL observed in our study in the active stage of the disease correspond to their high concentrations in skin.

It was observed that treatment decreases the total amount of AGEs in psoriatic patients [37]. The remission of psoriatic lesions in our study subjects was accompanied by significant decreases in the serum concentrations of CML and CEL. However, the concentrations during remission were still higher than in the healthy controls.

A relationship between CEL concentration and cumulative influence of therapeutical method and time to remission was observed among the patients in the remission phase. The multivariate analysis showed that patients receiving acitretin for more than 6 weeks exhibited serum concentration of CEL higher than any other analyzed group. The connection between CEL concentration and acitretin administration is hard to explain; however, it is known that some retinoids undergo photodegradation in vivo, which gives rise to methylglyoxal and glyoxal. Methylglyoxal is a direct substrate of CEL synthesis during glycoxidation [38].

Additionally, the results suggest that the chronic inflammation itself (regardless of the presence of visible lesions) is a source of intensified glycation/glycoxidation. Patients with initial high concentrations during the active stage exhibit relatively high concentrations during remission too. It may partially explain the observed lack of correlations between the PASI values and the concentrations of CML and CEL. It seems that CML and CEL are good markers of glycation in psoriasis, contrary to e.g., systemic lupus erythematosus [39].

Interestingly, the results pertaining sRAGE concentrations are contradictory to the data available in the literature. Previous studies noted a significant decrease in sRAGE concentrations in psoriatic patients, which additionally correlated negatively with PASI scores. However, our subjects are characterized by lower mean age and higher mean PASI values, when compared to these studies [36,40].

sRAGE is classically considered to be a protective factor against proinflammatory RAGE ligands, and, as such, is associated with defense against inflammation and oxidative stress. Low levels of sRAGE are believed to lead to overstimulation of pro-inflammatory signaling pathways in cells. Additionally, there is an alternative view on the role of sRAGE in the inflammatory processes. It is possible that persistent overstimulation of RAGE leads to its overexpression on the cell surface. Then sRAGE is produced as an alternative splicing variant of RAGE mRNA or as a product of RAGE cleavage. Thus, high levels of sRAGE would be rather a marker of intense inflammation than a protective factor, as it reflects tissue production of RAGE [41,42].

Our results support the latter view. A constant stimulation of RAGE via its ligands during psoriatic inflammation (especially during the active phase) could cause an enhanced expression of its gene. Then sRAGE may be produced from RAGE’s mRNA or cleaved from transmembrane RAGE, and its concentration in blood would therefore increase. During remission, the decrease in AGEs concentrations could cause a decrease in sRAGE concentration. However, no significant correlations between CML and sRAGE were observed in active psoriasis and during the remission. What is more, correlation between CEL and sRAGE was actually negative in the active phase of the disease.

Importantly, AGEs are not the exclusive ligands of RAGE. Among RAGE ligands, there are: molecules released from cells that are damaged or stressed (e.g., amyloid β peptide, S100 proteins, high-mobility group box 1 protein), cell adhesion molecules (e.g., macrophage-1 antigen), and molecules originating from infectious factors [41]. It is worth noting that CML and CEL did not correlate with sRAGE concentrations in the healthy controls. In conclusion, the interdependence between the concentrations of individual AGEs and the concentration of sRAGE seems to be much more complex.

Some studies report antioxidative role of sRAGE and describe inverse correlations between markers of oxidative stress and sRAGE levels in diabetes, atherosclerosis, and cardiometabolic disorders [43,44,45,46,47]. The presence of reactive oxygen species (ROS) and oxidative stress is a favorable condition for AGEs formation, as previously mentioned. The intensified formation of sRAGE ligands may deplete the receptor, resulting in the inverse correlation [47]. Additionally, activation of RAGE (due to low level of sRAGE) leads to inflammation and enhanced formation of ROS [45].

On the contrary, patients suffering from type II diabetes with retinopathy exhibit high levels of sRAGE and malondialdehyde when compared to patients suffering from type II diabetes without retinopathy and healthy subjects [42]. Metformin therapy reduces levels of sRAGE and oxidative stress markers in diabetic patients [48]. Additionally, high levels of sRAGE are related to a higher incidence of cardiovascular events and/or mortality in patients with [49] diabetes and coronary artery disease [50]. It is possible that in psoriasis the AGEs-RAGE-sRAGE axis is an example of a positive feedback loop too. Due to binding of AGEs to RAGE production of ROS increases and transcription factor NF-κB is permanently activated. Subsequently, NF-κB increases expression of genes of pro-inflammatory cytokines (e.g., IL1-β, TNF-α, VCAM-1) and of RAGE itself. Consequently, inflammatory response is amplified [41]. It is possible that only a successful therapy of psoriasis is able to break this vicious cycle (during remission concentrations of sRAGE in our patients decreased, but were still higher than in healthy controls).

## 5. Conclusions

In conclusion, psoriatic chronic inflammation is accompanied by an increase in blood concentrations of AGEs, such as CML and CEL, when compared to healthy control. These concentrations reflect the stage of the disease: they are higher during the active phase and they decrease during remission. Unfortunately, the AGEs concentrations in psoriatic patients are not comparable to healthy individuals even after the treatment resulting in remission. These data suggest that despite the therapy, the organs of psoriatic patients are permanently affected by the pathological process and intense glycoxidation. As of today, there are few studies regarding sRAGE role in psoriasis and their results are inconclusive. Presumably, sRAGE concentration cannot be considered a good general biomarker of the disease severity and anti-inflammatory potential. Our data and literature data suggest it acts differently in various diseases. It is unclear if soluble RAGE is released in response to AGEs and other ligands as a protective factor or is simply a by-product of RAGE overstimulation.

We hope that our work expands the knowledge about the role of AGEs-RAGE-sRAGE axis in various diseases, especially psoriasis, as there is not much known about the role of these ligands and receptors in psoriatic patients. In our opinion, these findings are clinically important, as the axis is a potential therapy target worth further investigation.

## Figures and Tables

**Figure 1 biomolecules-12-01870-f001:**
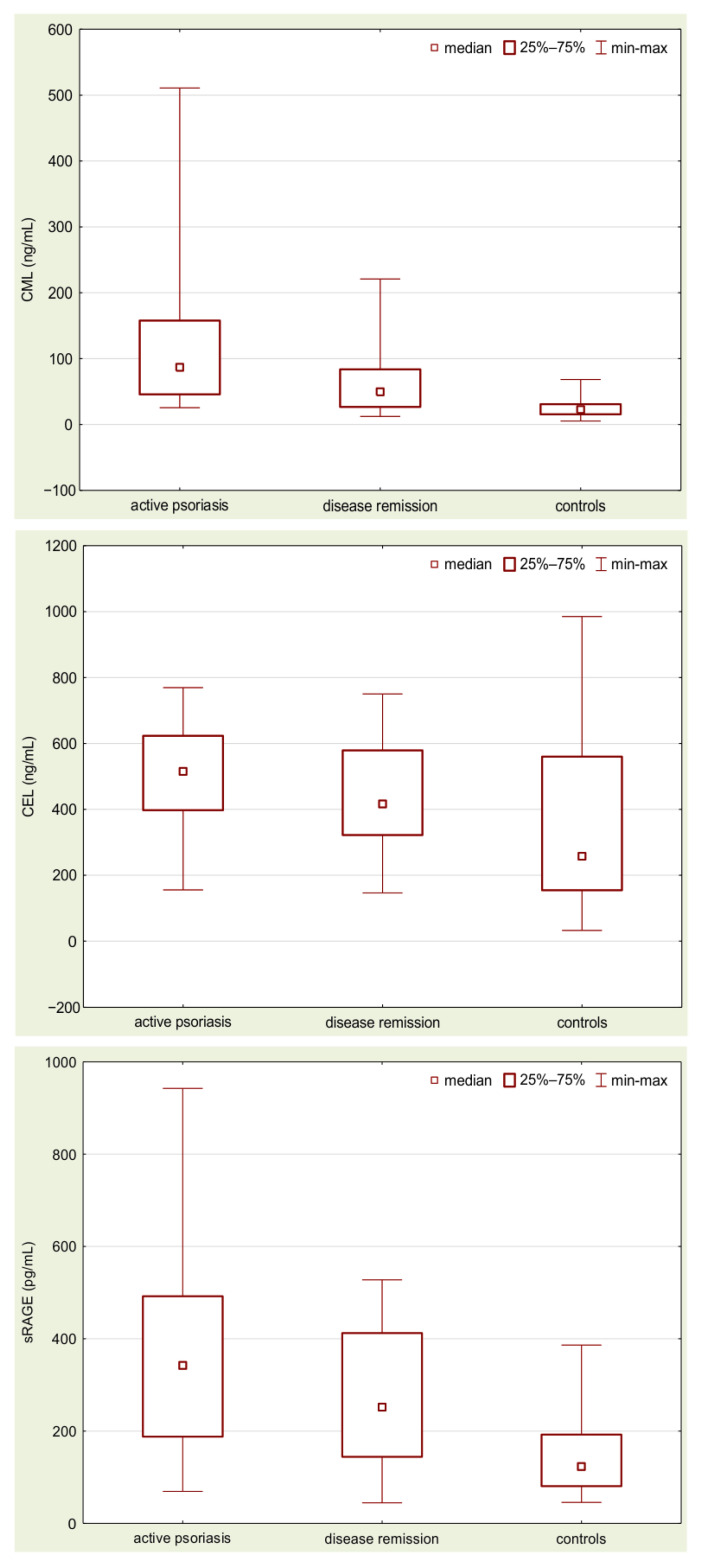
Concentrations of CML, CEL, and sRAGE in sera samples obtained from the patients with active disease, at the remission stage, and from the healthy controls.

**Figure 2 biomolecules-12-01870-f002:**
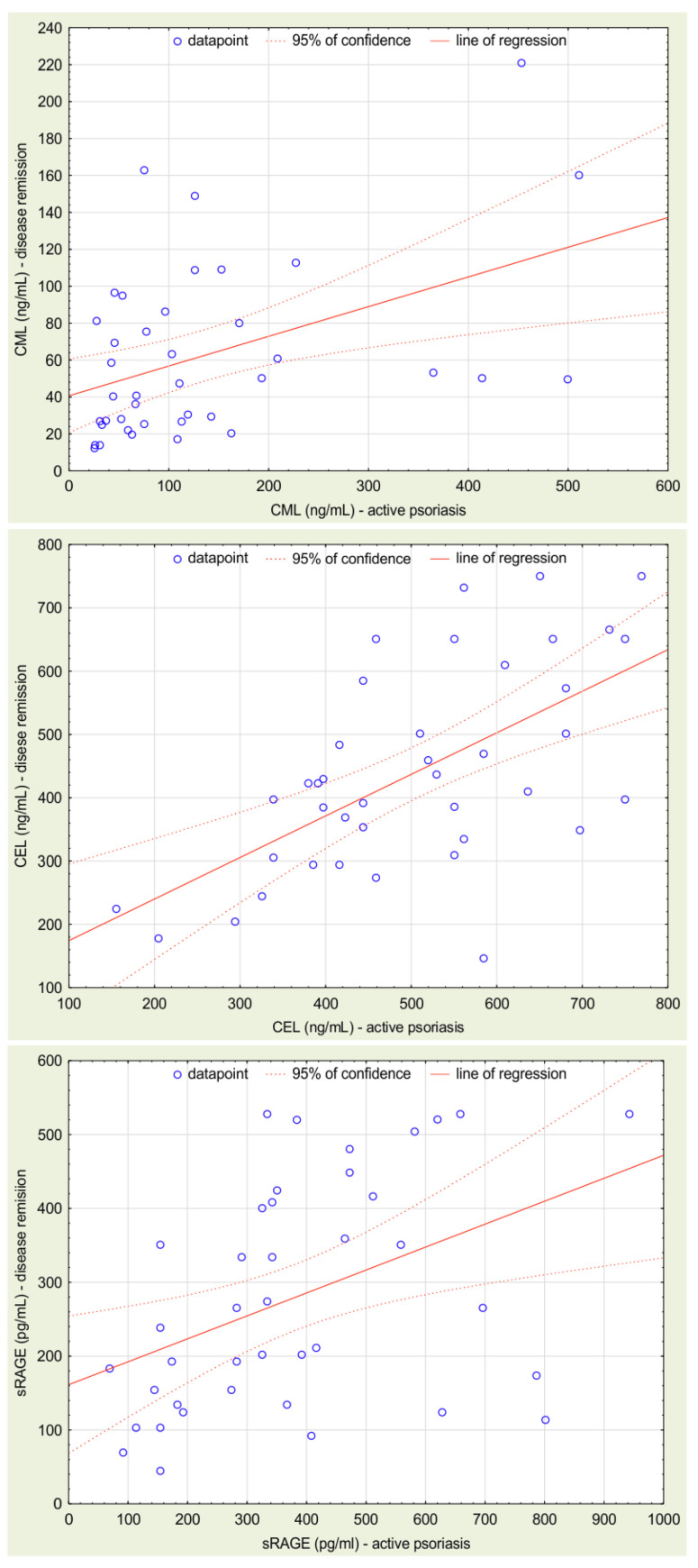
Correlations between CML, CEL, and sRAGE concentrations in sera samples obtained from the patients with active disease and from the patients at the remission stage.

**Table 1 biomolecules-12-01870-t001:** Clinical characteristics of patients.

	Active Phase Groups (*n*)		
PASI value	≤15	16–30	>30
	10	13	17
disease duration (years)	≤5	6–12	>12
	11	15	14
	**Remission Groups (*n*)**		
therapeutic method	topical corticosteroidsand UVB phototherapy	acitretin	ciclosporin
	11	14	15
time to remission (weeks)	0–3	4–6	>6
	15	15	10

**Table 2 biomolecules-12-01870-t002:** Concentrations of CML, CEL, and sRAGE in sera samples of the patients and healthy controls.

	Group		
Examined Parameters (Mean ± SD)	Psoriatic Patients (Active Phase) (*n* = 40)	Psoriatic Patients (Remission Phase) (*n* = 40)	Controls (*n* = 35)
CML (ng/mL)	134.64 ^a,b^ ± 132.45	62.35 ^a^ ± 47.92	24.22 ± 13.05
CEL (ng/mL)	506.07 ^a,b^ ± 151.04	440.97 ^a^ ± 162.56	340.70 ± 241.86
sRAGE (pg/mL)	380.81 ^a,b^ ± 211.93	279.55 ^a^ ± 153.09	142.26 ± 80.61

^a^ *p* < 0.05 psoriatic patients vs. healthy controls (U Mann–Whitney test). ^b^ *p* < 0.05 psoriatic patients (active phase) vs. psoriatic patients (remission phase) (Wilcoxon’s pair test).

**Table 3 biomolecules-12-01870-t003:** Correlations between the evaluated parameters.

	*R*	*p*
CML (active) vs. CML (remission)	0.45	0.0035
CEL (active) vs. CEL (remission)	0.58	0.00008
sRAGE (active) vs. sRAGE (remission)	0.47	0.0022
CML (active) vs. CEL (active)	0.16	0.39
CML (active) vs. sRAGE (active)	−0.11	0.51
CEL (active) vs. sRAGE (active)	−0.38	0.01
CML (remission) vs. CEL (remission)	−0.13	0.42
CML (remission) vs. sRAGE (remission)	0.16	0.32
CEL (remission) vs. sRAGE (remission)	−0.04	0.82
CML (controls) vs. CEL (controls)	−0.11	0.53
CML (controls) vs. sRAGE (controls)	−0.25	0.15
CEL (controls) vs. sRAGE (controls)	−0.15	0.38

## Data Availability

All obtained data are presented in the article.

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
