# Peer review of "Concentrations of N6-Carboxymethyllysine (CML), N6-Carboxyethyllysine (CEL), and Soluble Receptor for Advanced Glycation End-Products (sRAGE) Are Increased in Psoriatic Patients"

_biomolecules, 2022, doi:10.3390/biom12121870_

Round 1

Reviewer 1 Report

Manuscript "Concentrations of N6-carboxymethyl lysine ..." present an interesting clinical study dealing with analysis of glycation products and soluble RAGE receptor in patients with psoriasis vs healthy volunteers. The finding that the oxidative glycation products are elevated at psoriasis episodes and remain relatively high even at remission is important and clinically useful information. The manuscript is clearly written and is easy readable. One of few concerns that I have is (a) that statistical analysis on the data is not fully exhausting. With the data that the authors have they should also attempt more comprehensive statistical analysis using multivariate methods of data analysis. Additionally, (b) the authors do not comment on the treatment of psoriasis. Might be they could provide some treatment details (targeted mechanisms if different) and might be reasons if these treatments might disclose some correlations to the measurement data. (c) Time to remission is also one of the parameters that could be used in multivariate data analysis.

In conclusion, it seems that the data is not fully processed. More comprehensive analysis of data would elevate the value of this work.

Reviewer 2 Report

The Manuscript entitled "Concentrations of N6-carboxymethyl lysine (CML), N6-carboxyethyl lysine (CEL), and soluble receptor for advanced glycation end-products (sRAGE) are increased in psoriatic patients" discusses about the Psoriasis which is a chronic severe skin disease, associated with metabolic disorders and increased risk of cardiovascular complications. One of the reason for Psoriasis is advanced protein glycation and/or glycoxidation. Authors have assessed the concentrations of N6-carboxymethyl lysine (CML), N6-carboxyethyl lysine (CEL), and soluble form of receptor for advanced glycation end-products (sRAGE) in psoriasis patients at different phases of the disease activity, in comparison to healthy individuals. 

The reviewer is of the opinion that this MS can be considered for publication, however, the paper has to be improved as per the comments on point by point basis-

1. Overall check the spelling and the syntax and/or grammar error throughout the paper
2. The MS is found to have similarity with the previously published article, and it was recorded to be 30%. Authors are instructed to decrease the similarity percentage to the extent of less than 20%.
3. Page 2, Line 45, -- compound (for example monosaccharide, glyoxal, methylglyoxal) [ref. a]. Authors are advised to cite after the end of the sentence.

a. The role of advanced glycation end products in progression and complications of diabetes. The Journal of Clinical Endocrinology & Metabolism. 2008;93(4):1143-52.

3. Authors are advised to provide the detail methodology for ELISA for the estimation of CML, CEL, and s-RAGE.
4. Why there is inverse correlations between sRAGE and oxidative stress markers?
5. The conclusion should be written in a separate heading not to be merged with the discussion.
6. Discussion section: line no. 133. "AGEs are generated in vivo, under physiological and pathological conditions". There should be citation at the end of the sentence [b].

b. An insight on the association of glycation with hepatocellular carcinoma. Seminars in Cancer Biology 2018: 49, 56-63. 

7. Authors are advised to also write the significance of the study at the end of conclusion.

Round 2

Reviewer 2 Report

The Author's have replied to the comments raised by the reviewer. The revised version of the Manuscript can be accepted now.